# Trehalose and tardigrade CAHS proteins work synergistically to promote desiccation tolerance

Kenny Nguyen[1,3], Shraddha KC [1,3], Tyler Gonzalez[1], Hugo Tapia[2] & Thomas C. Boothby [1✉]

Tardigrades are microscopic animals renowned for their ability to survive extreme desiccation. Unlike many desiccation-tolerant organisms that accumulate high levels of the disaccharide trehalose to protect themselves during drying, tardigrades accumulate little or undetectable levels. Using comparative metabolomics, we find that despite being enriched at low levels, trehalose is a key biomarker distinguishing hydration states of tardigrades. In vitro, naturally occurring stoichiometries of trehalose and CAHS proteins, intrinsically disordered proteins with known protective capabilities, were found to produce synergistic protective effects during desiccation. In vivo, this synergistic interaction is required for robust CAHS-mediated protection. This demonstrates that trehalose acts not only as a protectant, but also as a synergistic cosolute. Beyond desiccation tolerance, our study provides insights into how the solution environment tunes intrinsically disordered proteins' functions, many of which are vital in biological contexts such as development and disease that are concomitant with large changes in intracellular chemistry.

[1] University of Wyoming, Department of Molecular Biology, Laramie, WY, USA. [2] California State University—Channel Islands, Biology Program, Camarillo, CA, USA. [3] These authors contributed equally: Kenny Nguyen, Shraddha KC. ✉email: tboothby@uwyo.edu

Water is required for all metabolic processes and is therefore often considered essential for life. However, a number of organisms across all biological kingdoms challenge this assertion by losing all, or nearly all, of the bulk water inside their bodies and cells and yet somehow surviving. For this to occur, these specialized organisms enter a state of suspended animation known as anhydrobiosis (Greek for 'life without water')[1–3]. Once in this anhydrobiotic state, dried organisms can persist for weeks, years, and in extreme cases centuries, and reactivate metabolism and resume life processes upon rehydration[1–3].

One such anhydrobiotic organism is the tardigrade, known colloquially as a water bear[4]. Tardigrades are microscopic, eight-legged, animals phylogenetically related to arthropods and nematodes[4]. There are currently ~1400 extant species of tardigrades that have been described, many of which are able to tolerate a number of abiotic stresses, including desiccation[5].

How tardigrades survive desiccation is one of the enduring mysteries of organismal physiology. Biochemical and molecular studies have begun to shed light on how tardigrades survive drying, and have implicated antioxidant enzymes and reactive oxygen species scavengers, conserved and tardigrade-specific disordered proteins, and DNA repair mechanisms in mediating desiccation tolerance[6–12]. These pioneering studies have been bolstered by the adoption of various -omics approaches, mainly stemming from the next-generation sequencing of genomes and transcriptomes of various tardigrade species[7,10,12–17]. While gene and transcript complements of tardigrades under desiccating conditions have been assessed to a degree, other -omics-level analyses, such as metabolomics, are lacking for tardigrades.

Understanding what changes to the metabolome of tardigrades take place during desiccation will likely be key in understanding how these animals survive desiccation. In other anhydrobiotic organisms, many small molecules and metabolites have been identified to have known or suspected roles in desiccation tolerance, ranging from vitrification to reactive oxygen species scavenging[18–20]. Bridging this gap in knowledge for tardigrades will help us better understand the commonalities and differences between how they and other desiccation tolerant organisms survive drying.

Here we address this knowledge gap by performing the first metabolome-wide assessment of changes in the small molecule/metabolite content of a tardigrade undergoing desiccation. We use the tardigrade Hypsibius exemplaris, as this species has been widely studied, is simple to culture to levels required for input for mass-spectrometry, and is anhydrobiotic when preconditioned[7,21,22]. This last point is important: since non-preconditioned animals do not survive drying, we can make direct comparisons between anhydrobiotic and non-anhydrobiotic animals of the same species.

We find that there are dramatic and reproducible changes to the metabolome of H. exemplaris as it dries. Several metabolites in the trehalose biosynthetic pathway, including trehalose, increase in enrichment in dry versus hydrated tardigrades and random forest analysis shows that trehalose and trehalose biosynthesis pathway components are important biomarkers distinguishing dry and hydrated H. exemplaris specimens. Trehalose is a known mediator of desiccation tolerance and is accumulated to high levels (~20% dry weights) in many anhydrobiotic organisms[23]. However, in H. exemplaris as well as all other tardigrade species examined to date, trehalose is accumulated to low or undetectable levels (0-2.9% dry weight)[24–26]. To address why trehalose is accumulated to such low levels in tardigrades, we test the sugar's ability to work synergistically with a known mediator of tardigrade desiccation tolerance, the tardigrade-specific disordered protein CAHS D. We find that when mixed with CAHS D, at or above the natural molar ratio, trehalose promotes a large and

significant synergistic increase in protection in our in vitro desiccation assay. Furthermore, trehalose does not promote a synergistic effect when mixed with bovine serum albumin (BSA), another desiccation protective protein[7,27] demonstrating that the interaction between CAHS D and trehalose is specific. Using a heterologous yeast system in which intracellular levels of trehalose can be controlled, we find that this synergistic effect extends to an in vivo setting as well as to other CAHS proteins, and that their combination with trehalose is required for CAHS proteins to exert their protective effect in this in vivo setting.

Together, our studies demonstrate that the metabolite content of the tardigrade H. exemplaris changes dramatically and reproducibly during desiccation and that at least one of these enriched solutes, trehalose, works synergistically with tardigrade CAHS proteins to promote in vitro and in vivo desiccation tolerance.

## Results and discussion

**Desiccation induces dramatic and reproducible changes to the metabolome of *Hypsibius exemplaris*.** In order to survive desiccation, the tardigrade Hypsibius exemplaris must first be 'preconditioned' by slow drying[7,10,12–17]. Essentially all H. exemplaris specimens survive preconditioning alone, ~89% survive preconditioning and drying, and none survive drying without preconditioning[7,10,12–17]. Thus, preconditioning likely elicits some change, possibly involving changes at the level of the metabolome, that help these animals cope with the stresses of desiccation. To begin to assess which metabolites might contribute to tardigrade desiccation tolerance we conducted global metabolomics on nine fully hydrated and nine preconditioned and desiccated samples of H. exemplaris (Fig. 1a). Both hydrated and desiccated animals were washed and starved prior to sample preparation to reduce possible contaminate from their algal food source. Preconditioned only animals were not tested due to the high number of animals needed for each condition.

Our analysis resulted in the identification of a total of 348 known biochemicals and 24 unnamed biochemicals (File S1). We performed differential enrichment analysis to identify biochemicals that are enriched or depleted under hydrated and dry conditions. A total of 271 biochemicals were found to be differentially enriched ($p \leq 0.05$) between hydrated and dry conditions (Fig. 1b, c). The majority of differentially enriched biochemicals (228 out of 271) were enriched in dry relative to hydrated conditions (Fig. 1b). In contrast, relatively few biochemicals were enriched in hydrated samples (43 out of 271) (Fig. 1b, c). These initial findings indicate that during drying the tardigrade metabolome changes relative to their hydrated metabolome, mostly through the accumulation of small molecules and metabolites.

To assess how reproducible changes to the metabolome are, we performed principal component analysis (PCA) on our 9 hydrated and 9 desiccated datasets. This analysis is used to reduce the dimensionality of the dataset by identifying contributors of variation based on their global metabolic profiles. PCA performed on our datasets strongly segregated hydrated from desiccated samples (Fig. 1d). This analysis suggests that not only does desiccation induce profound changes in the metabolite composition of tardigrades but that these changes are highly stereotyped such that all desiccated samples resemble each other more than they resemble any hydrated sample.

In addition to PCA, random forest analysis (RFA) was performed. This approach uses machine learning to sort, or bin, data points into two classes based on their dataset similarities. These class predictions are then compared to the true class of the samples, resulting in a prediction accuracy. RFA bins individual samples into groups based on their metabolite enrichment

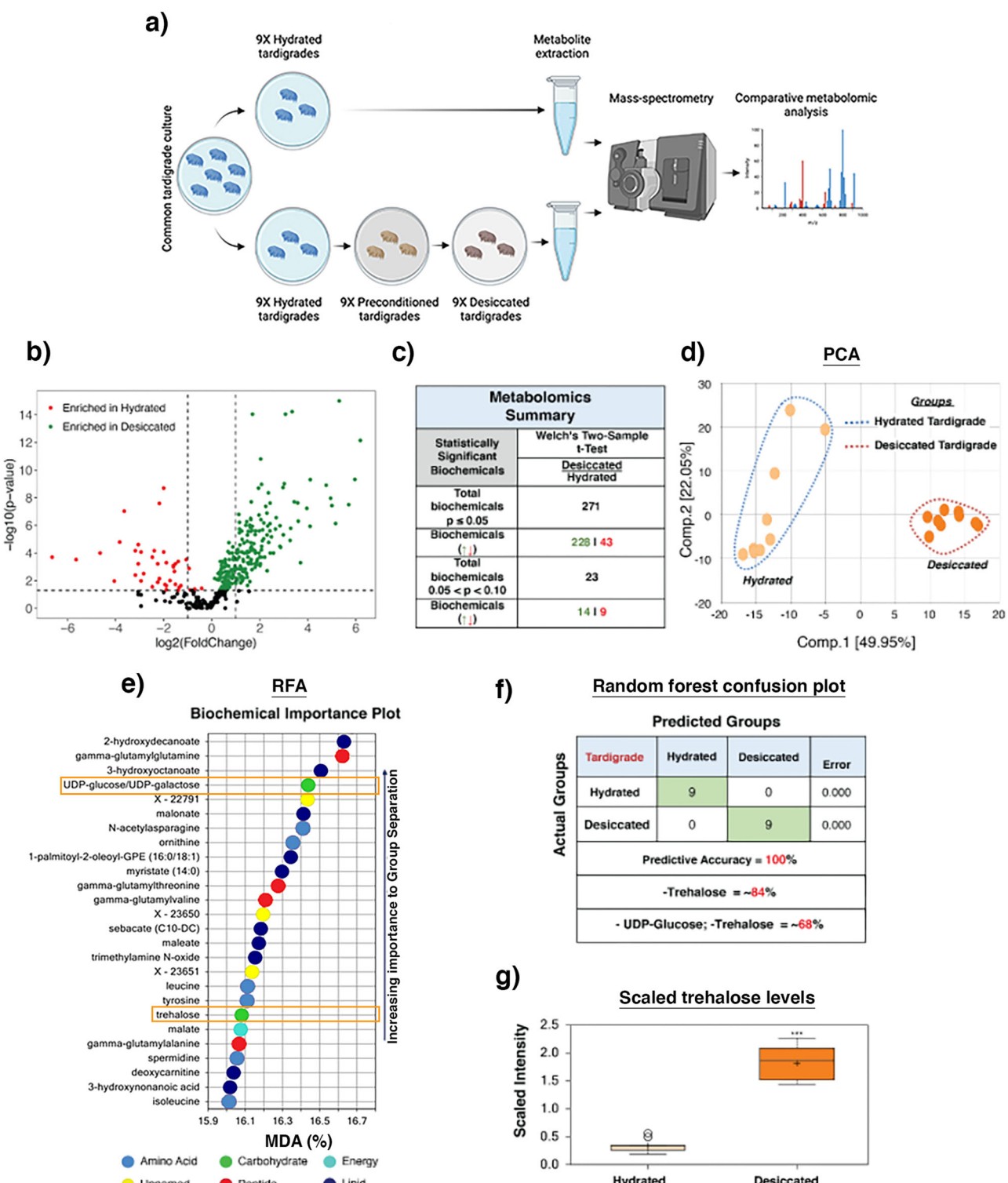

**Fig. 1 Metabolomic analysis of hydrated and desiccated tardigrades. a** Schematic representing preparations of nine hydrated and nine preconditioned and desiccated tardigrade samples for metabolomics analysis. **b** Volcano plot displays differential enrichment of metabolites with the change in hydration state of tardigrades. **c** Summary table of the metabolome profiling comparing hydrated and desiccated tardigrade samples. Welch's two-sample *t*-test used for statistical comparisons. **d** PCA showed major separation by hydration status on Component 1, suggesting that desiccation causes a substantial metabolic shift in tardigrades. **e** Random forest comparison between hydrated and desiccated groups in tardigrades identifies trehalose and trehalose precursor UDP-glucose as significant metabolites that contribute strongly to the group binning. The plot shows the mean decrease in binning accuracy that would result from the removal of a particular metabolite from our analysis. **f** RFA confusion matrix showing that predictive accuracy within our dataset is 100%. **g** Plot showing scaled intensity (relative metabolite abundance) for trehalose in hydrated and desiccated samples. $n = 9$, Welch's two-sample *t*-test was used for statistical comparisons where *** indicates *p*-value < 0.001, error bars = standard deviation.

similarities and differences. These computationally derived groups are then compared to the actual identity of the samples to derive predictive accuracy. Figure 1e shows RFA comparisons between hydrated and desiccated tardigrades, which resulted in a predictive accuracy of 100% (Fig. 1f). Since this accuracy would be expected to be 50% if binning was performed randomly, these results suggest that RFA was successful in binning samples to their appropriate groups.

RFA reports metabolites that contribute most strongly to the accuracy of the group binning. This analysis was done by omitting a metabolite and rerunning the modified dataset through the analysis to assess change in prediction accuracy. Mean decrease accuracy (MDA) was then calculated as the difference of prediction accuracy between the permuted sample and the true samples. This allowed us to identify metabolites that greatly affect the prediction accuracy when binning our hydrated and desiccated tardigrades samples. Metabolites that result in high MDAs can function as biomarkers for distinguishing between groups of interest. The top metabolites contributing to the binning of hydrated and dry tardigrade samples suggest a role in amino acid and peptide metabolism, changes in lipid composition, as well as unknown/uncharacterized metabolites in tardigrade desiccation tolerance (Fig. 1e). In addition, the disaccharide trehalose, which was found to be enriched in dry tardigrades ~5X compared to hydrated tardigrades (Fig. 1g), was observed to be a strong biomarker for distinguishing hydrated and dry tardigrades (Fig. 1e). Additionally, UDP-glucose which is an upstream component in the trehalose biosynthetic pathway was also identified by RFA as one of the top contributors to accurate binning. Trehalose and UDP-Glucose had MDA scores of ~16% each (Fig. 1e), meaning that the removal of either of these metabolites from our RFA analysis results in a decrease from 100% predictive accuracy to ~84% predictive accuracy. Likewise, removal of both trehalose pathway metabolites results in a decrease from 100% accuracy to ~68% accuracy (Fig. 1f).

Combined, our metabolomic profiling of hydrated and desiccated *H. exemplaris* specimens shows that changes to the metabolome of this tardigrade species during drying are dramatic and reproducible. Furthermore, a number of biochemical pathways and types of metabolites are modulated during desiccation, including components of the trehalose biosynthetic pathway, which serve as strong biomarkers for the identification of dry *versus* hydrated specimens.

**Trehalose metabolism and accumulation in *Hypsibius exemplaris*.** While several metabolites had slightly higher mean decrease accuracy scores than trehalose and UDP-glucose, we chose to pursue examining the role of trehalose in tardigrades because, trehalose is a known mediator of natural desiccation tolerance in a number of anhydrobiotic organisms and can confer protection against drying in a number of heterologous in vitro, ex vivo, and in vivo systems[28–31]. Furthermore, despite the long, well-documented history of trehalose promoting desiccation tolerance in myriad systems, the role of this sugar in tardigrade anhydrobiosis is enigmatic[24–26].

Whereas trehalose is found to accumulate to high levels, such as ~20% dry weight in yeast, brine shrimp cysts, *Polypedilum vanderplanki*, and the dauer larva of *Caenorhabditis elegans*, in tardigrades previous biochemical analysis reveals that trehalose is accumulated at low abundance (>0.1–2.9% dry weight) in some species or completely undetectable in others[24–26]. The presence of detectable trehalose in tardigrades appears to be species dependent, suggesting a possible divergence in anhydrobiotic survival strategies[25,26]. Similarly, statistically significant accumulation of trehalose upon desiccation in tardigrades appear to be

species specific, with some species seeing appreciable differences in trehalose levels between hydrated and anhydrobiotic specimens while in other species there were no detectable differences in trehalose levels between hydrated and dry conditions[24–26]. The relatively low levels, or in some cases complete absence, of detectable trehalose and the varying degrees to which trehalose is enriched during desiccation in tardigrades has made some question this sugar's role in tardigrade anhydrobiosis.

Adding to the ambiguity over the use of trehalose in tardigrade desiccation are genomic and transcriptomic studies of the genes involved in trehalose synthesis. The canonical trehalose biosynthetic pathway relies on the glycosyl-transferase, trehalose-6-phosphate synthase (TPS). The gene (*tps*) encoding the TPS enzyme is found in many animals, including ecdysozoans closely related to tardigrades such as the nematode worm *Caenorhabditis elegans* and the fruit fly *Drosophila melanogaster*. The first study assessing the presence of *tps* in tardigrades examined 12 phylogenetically distributed species[32] and found that all semi-terrestrial and anhydrobiotic species examined possessed *tps*. In contrast, *tps* was not detected in two freshwater and desiccation-sensitive species, however a third freshwater, desiccation-sensitive, species did possess *tps*[32]. Subsequent genomic and transcriptomic studies support the notion that trehalose and trehalose synthetic pathway genes and their expression vary between tardigrade species[12,16,17]. A recent report suggesting that recurrent loss of endogenous genes followed by horizontal gene transfer of genes involved in trehalose biosynthesis in anhydrobiotic organisms may account for the discrepancies observed in the presence of trehalose-related genes in different tardigrade species[33].

Considering the enigmatic nature of trehalose's involvement in tardigrade anhydrobiosis, we were intrigued by the enrichment and appearance of trehalose, and trehalose pathway products, in our RFA (Fig. 1e). Trehalose's inclusion in this list indicates that beyond present in dry specimens, the enrichment of this sugar is a key metabolomic biomarker distinguishing desiccated from hydrated *H. exemplaris*. This suggests that at least in *H. exemplaris*, trehalose may be important for anhydrobiosis.

Since trehalose is typically seen to accumulate to high levels in non-tardigrade anhydrobionts, but not in tardigrades, we quantified the amount of trehalose in dry *H. exemplaris* (Table 1 and Fig. S1). Approximately, 85 mg of preconditioned and dried *H. exemplaris* animals (~260,000 animals) were homogenized and analyzed to assess total trehalose content. 1.6 µg of trehalose (0.0019% dry weight) was detected in this sample (Table 1 and Fig. S1). This brings the amount of trehalose in a single dried *H. exemplaris* tardigrade to ~0.0061 ng, or ~0.016 pmol per tardigrade (Table 1). These measurements are in line with previous reports that indicate the dry mass of trehalose in tardigrades typically ranges from 0 to 0.47% or in extreme cases as high as 2.9% depending on the species of tardigrade under study[24–26].

To gain insight into why there are relatively low levels of trehalose in anhydrobiotic *H. exemplaris* specimens, we assessed the presence and level of gene expression for key components of the trehalose biosynthetic pathway in *H. exemplaris* under hydrated and desiccated conditions. Figure 2 summarizes the results of this analysis. All key enzymes involved in trehalose biosynthesis were identified in *H. exemplaris* (Fig. 2). Homologs of upstream biosynthetic enzymes (Hex-A, PGM1, UGP) displayed variable expression patterns with some homologs upregulated during desiccation and others downregulated. Genes encoding trehalase (Treh), the enzyme responsible for breaking down trehalose, were all downregulated under drying conditions (Fig. 2). Genes encoding TPS1 and TPP, the last two enzymes in the trehalose biosynthetic pathway were found to be present in

**Table 1 Quantification of trehalose and putative CAHS protein levels in *H. exemplaris*.**

| Protectant | Number of tardigrades | Quantity (µg) | Moles of protectant (nmol) | Quantity per tardigrade (ng) | Moles of protectant per tardigrade (pmol) | Concentration (mg/ml) | Molar ratio trehalose:CAHS | % dry mass per tardigrade |
|---|---|---|---|---|---|---|---|---|
| Trehalose | ~260,000 | 1.6 | 4.23 | 0.0061 | 0.016 | 0.134 | ~8:1 | 0.0019% |
| Putative CAHS | ~260,000 | 13.1 | 0.51 | 0.050 | 0.0019 | 1.10 | ~8:1 | 0.015% |

Summary table displaying quantitative values of detectable trehalose (Fig. S1) and CAHS protein (Fig. S2) in desiccated tardigrades.

the transcriptome of *H. exemplaris* but were found to not be expressed at appreciable levels under either hydrated or dry conditions (Fig. 2). These data suggest that during drying of *H. exemplaris*, there is not a strong enrichment in gene products promoting the biosynthesis of trehalose (e.g., Tps1 or TPP), but there is a systematic down regulation of gene products involved in the breakdown of trehalose (trehalases). This suggests that *H. exemplaris* may have evolved to enrich low levels of trehalose during drying, not through the synthesis of more of this sugar, but rather via the inhibition of the sugar's breakdown. Further studies looking at the enzyme activities of trehalases in drying tardigrades could help support this idea.

Combined data from metabolomic, enzymatic, and transcriptomic analysis of trehalose and genes involved in the trehalose metabolism indicate that while trehalose is not accumulated to the high-levels seen in some other anhydrobiotic organisms, the relatively low-level of trehalose observed to accumulate in desiccated *H. exemplaris* specimens is none-the-less statistically significant and a key biomarker distinguishing hydrated and dry *H. exemplaris* specimens.

**In vitro synergy between trehalose and the tardigrade intrinsically disordered protein CAHS D**. The relatively low levels of trehalose observed in anhydrobiotic *H. exemplaris* specimens in our current study, as well as in a number of other tardigrade species in previous investigations[24–26], led us to wonder whether trehalose might not be conferring a large degree of protection to drying tardigrades by itself, but rather might work synergistically with another protectant(s) to confer protection. Previous studies have shown that trehalose and heat shock proteins (HSP) can work synergistically both in vitro and in vivo to promote desiccation tolerance[34,35]. If low levels of trehalose are working synergistically in *H. exemplaris* with another protectant, we reasoned tardigrade cytoplasmic abundant heat soluble (CAHS) proteins could potentially be synergistic partners.

CAHS proteins are a tardigrade-specific family of intrinsically disordered proteins (IDPs) that have previously been shown to confer desiccation tolerance both in vivo and in vitro[7]. IDPs lack a stable three-dimensional structure and instead exist in a series of interconverting conformations known as an ensemble. An IDP's ensemble, much like the structure of a well-folded protein, is thought in large part to drive its function(s)[36,37]. However, due to IDPs' lack of intramolecular bonds and large solvent exposed surface areas, their ensembles (and presumably functions) are known to be influenced by the solution environment[38–40]. Therefore, accumulation of trehalose, even at low levels, might influence the ensemble and protective capacity of CAHS proteins.

While previous studies have demonstrated synergism between trehalose and HSPs, these experiments were mixed at non-biological ratios of trehalose and HSPs[34,35]. Unfortunately, due to CAHS proteins' high degree of disorder, antibodies have not been raised against these proteins, making quantification of their levels via western blot or a similar approach impossible. With this in mind, we obtained an approximate biological molar ratio of trehalose to CAHS protein in dry *H. exemplaris* specimens taking advantage of the fact that CAHS proteins are highly heat soluble. The heat solubility of CAHS proteins means that when boiled they do not aggregate and precipitate out of solution. Taking advantage of this fact, we homogenized lysates of desiccated tardigrades, boiled their lysates which causes their non-heat soluble proteins to crash out of solution, and analyzed the heat soluble fraction containing CAHS proteins by SDS-PAGE gel (Fig. S2). A band corresponding to ~26 kDa, the approximate size of most CAHS proteins, was quantified by comparison to a series of bovine serum albumin standards (Fig. S2). It should be noted

## Trehalose Biosynthesis Pathway

| comp_ID | logFC | |
|---|---|---|
| comp84042_c0 | 1.39 | ↓ |
| comp84042_c1 | 1.52 | ↓ |
| comp73179_c0 | -1.66 | ↑ |

| comp_ID | logFC | |
|---|---|---|
| comp90923_c0 | 0.86 | ↓ |
| comp91603_c0 | 1.03 | ↓ |
| comp88742_c0 | -1.14 | ↑ |

| comp_ID | logFC | |
|---|---|---|
| comp86246_c0 | 1.80 | ↓ |
| comp86763_c0 | 2.06 | ↓ |

| comp_ID | logFC |
|---|---|
| comp67137_c0 | - |
| comp145573_c0 | - |
| comp169665_c0 | - |

| comp_ID | logFC |
|---|---|
| comp119539_c0 | - |
| comp161562_c0 | - |
| comp140955_c0 | - |

| comp_ID | logFC | |
|---|---|---|
| comp85602_c0 | 0.90 | ↓ |
| comp88604_c0 | 1.49 | ↓ |
| comp88604_c1 | 1.23 | ↓ |

**Fig. 2 Reciprocal best BLAST analysis results for trehalose pathway components in *H. exemplaris*.** Orthologs of tardigrade biosynthesis genes identified through reciprocal best BLAST analysis mapped to differential gene expression. Comp_IDs indicate transcript accession numbers. LogFC indicates the log2 fold change values for each transcript in a hydrated state relative to the desiccated state (such that a negative number indicates a higher enrichment in the dry state).

that this analysis is not specific or encompassing for all CAHS proteins, with some CAHS protein likely existing in other bands that were not quantified, or conversely it is possible that other proteins of ~26 kDa were also heat soluble and included in the analysis. So rather than an absolute quantification, this result gives us a rough estimate of the ratio of CAHS protein and trehalose in dry *H. exemplaris* specimens.

This analysis indicates that there is ~0.050 ng, or 0.0019 pmol, of putative CAHS protein per tardigrade (Table 1 and Fig. S2). Combining this number with our assessment of trehalose levels in dried *H. exemplaris* specimens (Table 1) we obtained a biological molar ratio of ~8:1 (trehalose:CAHS) (Table 1). As noted above, this ratio is likely a lower limit estimate of the ratio of trehalose to CAHS protein, and the actual ratio of trehalose to CAHS protein, and certainly the ratio of trehalose to a specific (e.g., CAHS D) CAHS protein, is likely higher.

To test for synergy between trehalose and CAHS proteins, we tested the protective function of HeCAHS D (CAHS 94063: UniRef100_P0CU45) and trehalose in an in vitro lactate dehydrogenase (LDH) desiccation assay (Fig. 3; Fig. 4). LDH is an enzyme that is known to be particularly sensitive to desiccation, with drying and rehydrating reducing the function-ality of the enzyme to ~1–5% its original activity. This assay is widely used within the desiccation tolerance field to assay for protective capacity of desiccation protectants[7,27,41]. We chose CAHS D as a model CAHS protein for this in vitro assay since to date, CAHS D is the most widely characterized CAHS protein, has been shown to provide protection to enzymes during drying in vitro, is essential for robust survival of *H. exemplaris* during desiccation, and improves the desiccation tolerance of hetero-logous in vivo systems[7,27]

In addition to CAHS D and trehalose, we assessed the protective capacities of bovine serum albumin (BSA) and sucrose in our LDH assay (Fig. 3; Fig. 4). BSA was chosen since it is a well-folded protein whose human homolog, human serum albumin, is used as an excipient by the pharmaceutical industry, and BSA itself is known to confer protection to enzymes during desiccation[7,27]. BSA here serves as a control, since BSA should: (i) not be strongly influenced by its solution environment, owing to

its well-folded nature, and (ii) not have coevolved with trehalose to confer desiccation tolerance. Sucrose was chosen as a control for trehalose as both are non-reducing disaccharides. Sucrose has known roles in plant desiccation tolerance[42–44], but is not produced by animals including tardigrades. At the molecular level, the major difference between trehalose and sucrose is that trehalose is composed of two glucose molecules joined by a 1–1 alpha bond, while sucrose is composed of a glucose and fructose joined by an 1–2 ether bond (Fig. 3b, c). Concentration dependent protection was observed to be conferred by CAHS D, trehalose, BSA, and sucrose in our LDH assays to varying degrees (Fig. 3). Other metabolites that were highly enriched in our metabolomics study were also tested in our LDH assay individually. Of the metabolites studied, none were able to confer significant protection independently besides trehalose (Fig. S3).

Concentration dependence protection data (Fig. 3a) was used to select suboptimal concentrations of protectants for protection assays with mixtures of a protein and sugar protectant. Selecting suboptimal concentrations allows us to test for synergy without the fear of reaching a potential 100% protective additive effect, for which additive or synergistic effects would be impossible to distinguish. CAHS D and BSA were then mixed with trehalose or sucrose at our empirically determined biological ratio (8:1) as well as an order of magnitude below (0.8:1) and above (80:1) this ratio. It should be noted that because CAHS D and BSA provide different degrees of protection, more BSA was needed to achieve similar protective effects in our synergy assays. As a result, to maintain the same ratios of trehalose:protein, more trehalose was also used in BSA mixing experiments.

When trehalose was mixed with CAHS D below the biologically determined ratio (0.8:1) no increase in protection beyond what would be expected from additive protection was observed (Fig. 4a). At and above the biological ratio of CAHS D to trehalose (8:1 and 80:1, respectively) synergistic protection was observed at levels statistically beyond what would be expected due to additive effects alone (Fig. 4a). By contrast, CAHS D did not begin working synergistically with sucrose until mixed at an order of magnitude higher than with trehalose (Fig. 4b). Interestingly, mixtures of BSA and trehalose or BSA and sucrose conferred

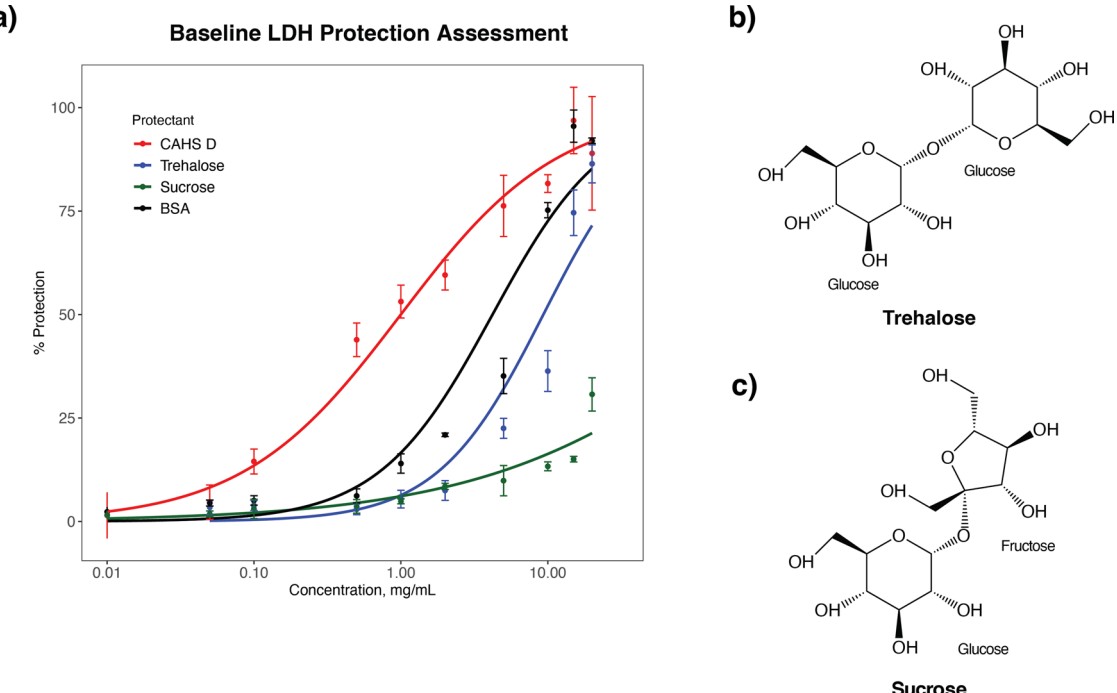

**Fig. 3 Lactate dehydrogenase protection assay baseline values for CAHS D, trehalose, sucrose, and BSA. a** Plot showing the percent protection of LDH conferred by each protectant as a function of concentration (mg/mL). $n = 3$, Error bars = standard deviation. **b** The molecular structure of trehalose. **c** The molecular structure of sucrose.

additive, but not synergistic protection in both assays at all molar ratios tested (Fig. 4c, d). These experiments demonstrate that CAHS D and trehalose mixtures work synergistically at and above biologically relevant ratios to confer protection to enzymes during desiccation. Additionally, CAHS D appears to have some specific preference for trehalose, as an order of magnitude higher levels of sucrose were required before synergy was observed. Importantly, the synergism observed between our sugar protectants, trehalose and sucrose, did not extend to mixtures including the well-folded protein BSA, suggesting that these sugars exert more of an effect on IDPs than globular proteins.

To assess whether trehalose and CAHS D confer synergistic protection to other biomolecules beyond LDH, we tested the protective capacity of each alone and in mixtures in an RNA integrity assay. mRNA encoding GFP was desiccated alone, with CAHS D, trehalose, or mixtures of CAHS D and trehalose at our three molar ratios. Desiccated RNA was rehydrated and the integrity of RNA assessed by using it as input in an in vitro translation assay. Protection was inferred by measuring the normalized fluorescence resulting from the translation of GFP.

Trehalose and CAHS D alone conferred no protection to desiccated RNA, except for trehalose at the highest concentration (Fig. S4). However, the effect of mixing trehalose and CAHS D was observed to be antagonistic at all molar ratios tested (Fig. S4). Combined with the results from our LDH protection assay, these data suggest that the synergistic effect observed between trehalose and CAHS D does not extend to all types of protection and may be specific for the stabilization of proteins during desiccation.

**In vivo synergism between trehalose and CAHS proteins**. Admittedly, the environment encountered by CAHS D in our in vitro assays (trehalose in buffer) is far from a true recapitulation of an anhydrobiotic cell. To assess whether the synergy observed between trehalose and CAHS D extends to an in vivo setting as well as to other CAHS proteins, five strains of

*Saccharomyces cerevisiae* each expressing a different tardigrade CAHS protein were generated. These strains were engineered in a *nth1Δ + TDH3pr-Agt1* background. (Fig. 5a). Yeast with this background lack Nth1, an essential gene required for the intracellular degradation of trehalose. Additionally, yeast with this background constitutively express Agt1, an alpha-glucoside transporter previously shown to efficiently import trehalose supplemented in yeast growth media[7]. Finally, yeast in their log growth phase do not express Tps1 *or* Tps2, enzymes required for trehalose production. As a result, yeast with this background growing logarithmically do not produce trehalose, or break it down, but can be loaded with trehalose through supplementation of the disaccharide in their media (Fig. 5a). This allows us to assess the protective effect of CAHS proteins and trehalose alone or in combination using these strains.

Yeast with the *nth1Δ + TDH3pr-Agt1* background containing an empty vector (no CAHS protein) grown in media not supplemented with trehalose displayed almost no desiccation tolerance (Fig. 5b–f). In three of five instances, expressing a single CAHS protein alone without any trehalose supplementation provided modest, but statistically significant, protection (Fig. 5b–f). Strains containing an empty vector grown in media supplemented with 2% w/v trehalose showed increased survival, confirming previous reports that trehalose is sufficient for conferring desiccation tolerance in yeast[30]. For all five CAHS genes tested, strains expressing a single CAHS protein and supplemented with 2% trehalose showed robust synergistic levels of survival, well above what would be expected if CAHS proteins and trehalose merely worked additively to promote anhydrobiosis (Fig. 5b–f). These data demonstrate that the synergistic effects of trehalose and CAHS D observed in vitro can extend to an in vivo setting and that this synergism extends to other CAHS proteins beyond CAHS D. Furthermore, these results suggest that in vivo CAHS-trehalose interactions may be an example of *obligatory synergism*, where in order to achieve robust protection, CAHS proteins require the presence of trehalose.

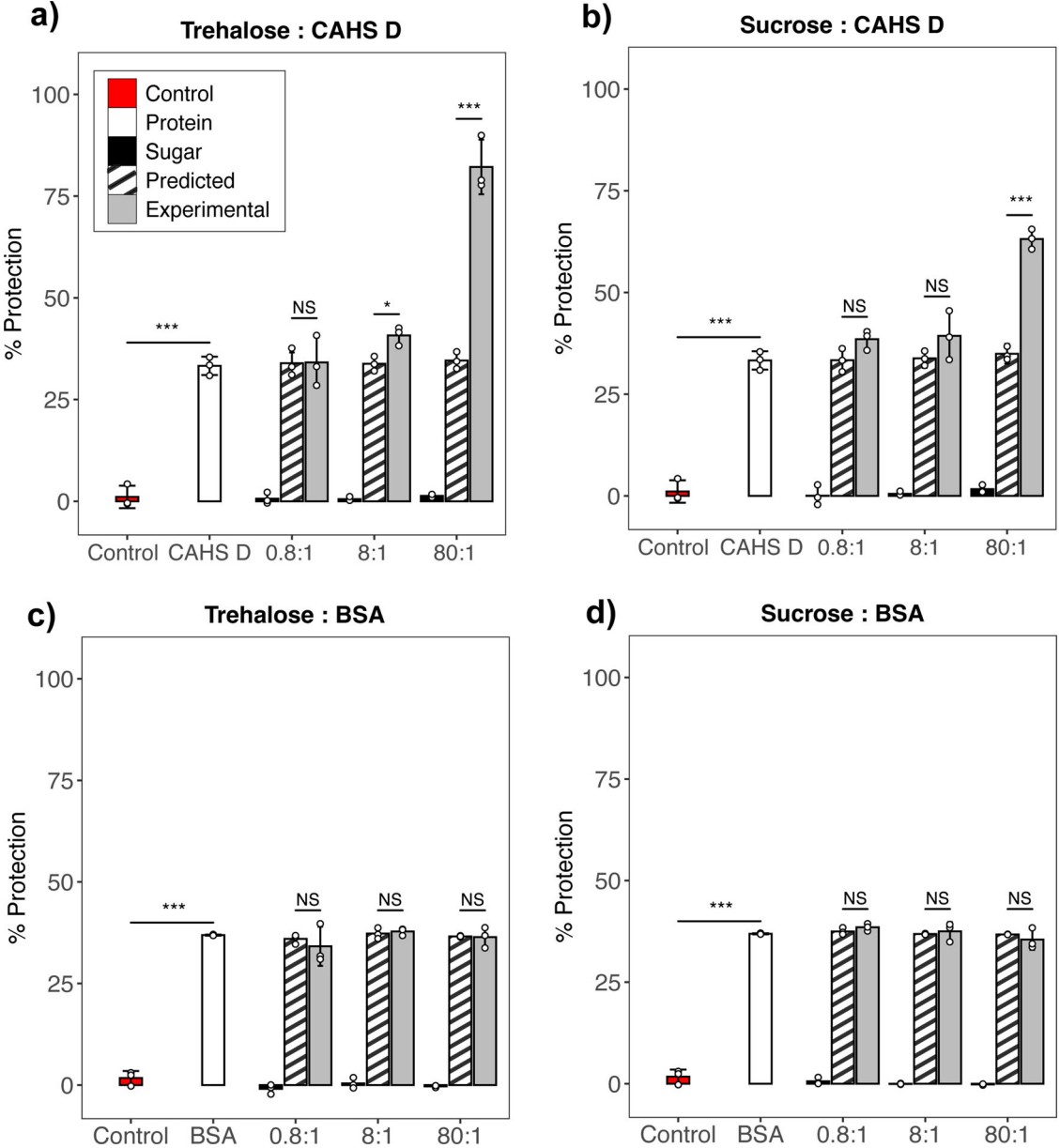

**Fig. 4 Trehalose and CAHS protein confers synergistic protection at and above estimated biological ratios.** Red bars represent negative controls in which no protectants were added to LDH solutions before drying. White bars represent samples in which only CAHS D was added to LDH solutions prior to drying. Black bars represent the percent protection of LDH conferred by either trehalose or sucrose. Striped bars represent the predicted additive of the sugar and the protectant (white + black bars). Gray bar represents actual experimental protection resulting from sugar and protein mixtures. Samples assayed were **a** Trehalose and CAHS D, **b** Sucrose and CAHS D, **c** Trehalose and BSA, **d** Sucrose and BSA. $n = 3$, one way ANOVA was used for statistical comparisons where ***, ** and * indicates $p$-value of <0.001, <0.01, and <0.05, respectively. Error bars = standard deviation.

Here we have examined changes to the metabolome of tardigrades that manifest upon desiccation. We find a number of biochemicals, and metabolic pathways are enriched upon drying, with statistical analysis showing that the enrichment of trehalose is important in distinguishing dry from hydrated tardigrades. Despite this, trehalose is present at much lower levels in tardigrades compared to other desiccation tolerant organisms that utilize this sugar to survive drying. Consistent with previous reports on other tardigrade species, we have shown that *H. exemplaris* possesses low levels of trehalose. Furthermore, this sugar, when mixed at biologically relevant ratios with CAHS D exerts synergistic protective effects. Importantly, this synergistic effect is not found between trehalose and the desiccation protective, well-folded protein, BSA. This synergism observed

between CAHS D and trehalose manifests in an in vivo setting and extends to other CAHS proteins, and interestingly in this setting trehalose is required for robust CAHS D mediated protection, indicating that synergy in this case may be obligatory.

These results demonstrate that small molecules enriched during the drying of anhydrobiotic organisms may not always serve as protectants in and of themselves but may contribute to desiccation tolerance by working together with other mediators. Furthermore, these findings illuminate a long-standing question in tardigrade biology: Does trehalose contribute to tardigrade desiccation tolerance and if so, why do these robust anhydrobionts make trehalose at such low levels relative to other organisms that survive drying?

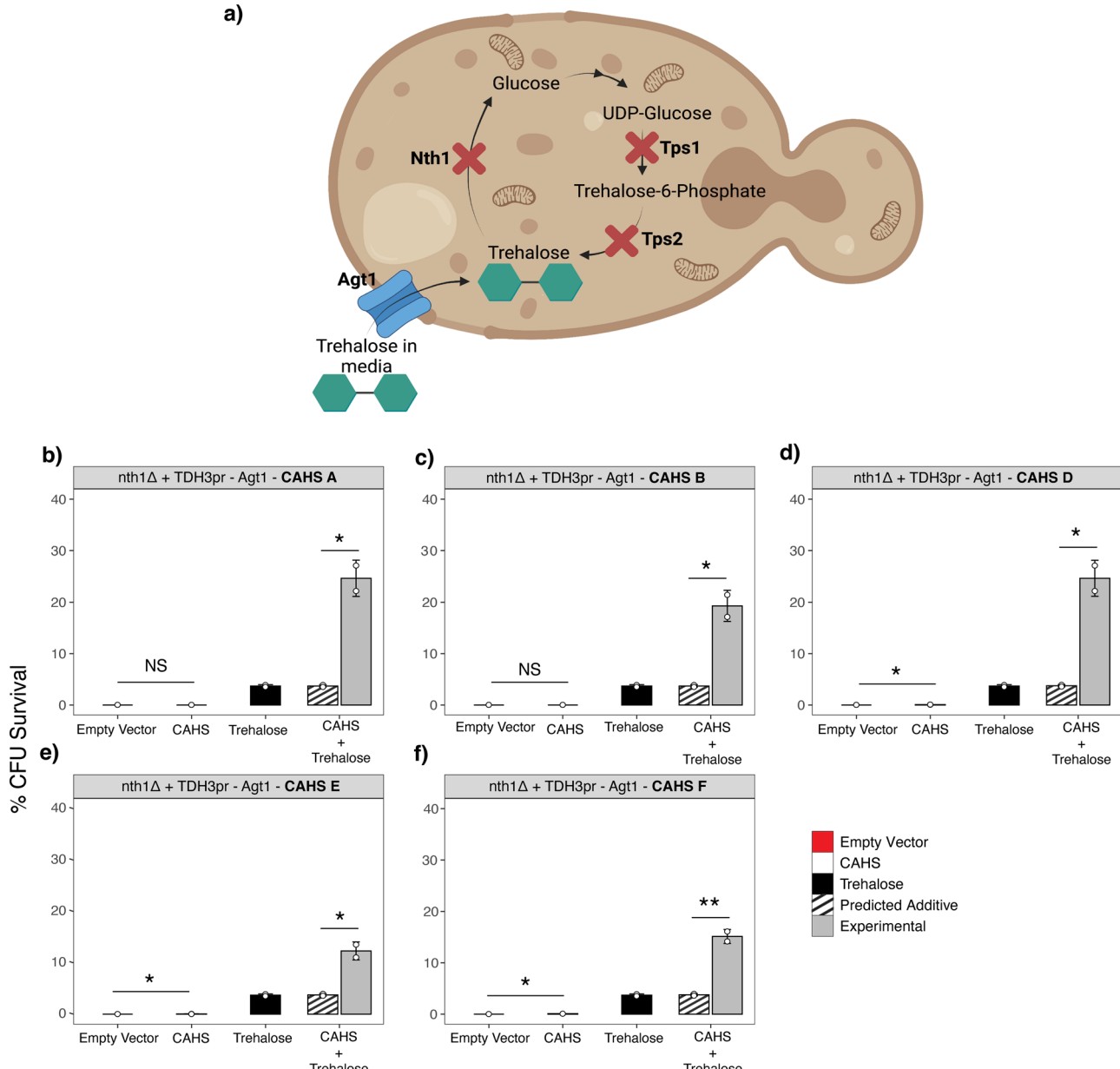

**Fig. 5 In vivo yeast synergy assay. a** Yeast strain *nth1Δ + TDH3pr-Agt1* was constructed to have a knockout of Nth1 which encodes trehalase for trehalose breakdown, and the addition of an intracellular trehalose transporter, AGT1. Yeast grown in log phase do not express Tps1 or Tps2. **b–f** *nth1Δ + TDH3pr-Agt1* yeast strains expressing different CAHS genes with and without supplemented trehalose were assayed for survivability against desiccation stress. One way ANOVA was used for statistical comparisons where *** indicates *p*-value < 0.001, error bars = standard deviation.

A long-term goal of the desiccation tolerance field is to understand better how to confer anhydrobiotic abilities to organisms that do not naturally survive drying. This study and its findings provide a compelling argument that to do so may require the combination of different, synergistic, protectants. This point is highlighted by our yeast desiccation assays (Fig. 5) where CAHS proteins confer little protection when heterologously expressed alone but provide dramatically increased protection in the presence of trehalose. Further understanding the mechanistic basis of CAHS-trehalose synergy will help to build a foundation for pursuing the long-term goal of engineering desiccation tolerance into organisms for the purposes of crop development and food security.

Beyond extending our knowledge of tardigrade physiology and desiccation tolerance, our finding that trehalose can modulate the protective capacity of CAHS proteins provides a compelling empirical demonstration that changing cellular, or solution chemistries can modulate the function of an IDP. This phenomenon likely extends beyond CAHS proteins and desiccation to IDPs more generally.

## Methods

**Tardigrade culture and concentration for metabolomics**. *H. exemplaris* specimens were obtained from Sciento (UK) and cultured under established conditions[21]. To concentrate tardigrades and remove them from their algal food source for metabolomic processing, tardigrades cultures were passed through a series of filters (200 mm, 100 mm, 75 mm, 50 mm, 30 mm) and backwashed off the 50 mm filter, which catches the majority of animals but allows their algal food to pass through, into a 50 mL conical tube.

To establish the density of tardigrades in our filtered stock, three 10 mL aliquots were taken and placed on a slide. Tardigrades in each aliquot were counted under a dissecting light microscope and an average density obtained. Using this average density, ~1700 tardigrades were transferred to eighteen 1.5 mL Eppendorf tubes. Nine of these aliquots were kept in a starved hydrated state, while the other nine

were starved and then desiccated according to standard procedures[7,45]. Samples were starved to avoid contamination from their algal food source. Hydrated and desiccated samples were submitted to Metabolon (Morrisville, NC) for metabolomic assessment.

**Metabolomics study to compare metabolome profile of hydrated and dried tardigrades**. Metabolomic study was conducted by Metabolon, Inc (Morrisville, NC). All procedures were carried out using their standard protocols and procedures as detailed in supplemental information. For scaled intensity measurements (e.g., Fig. 1g) raw scores from mass spectrometry were normalized for each replicate such that the median value was equal to 1. These normalized scores were further adjusted to account for differences between replicate BCA protein levels (protein normalization) and these protein normalized values were again set so that the median value was equal to 1.

**Reciprocal BLAST search to identify trehalose-biosynthesis orthologs**. Transcriptomic assembly and differential gene expression datasets for hydrated and dried tardigrades were obtained from Boothby et al.[7,45]. Protein sequences from *D. melanogaster* involved in trehalose biosynthesis were obtained from http://www.flybase.org and BLAST analysis was performed against the *H. exemplaris* transcriptome with an E-value cutoff of 1E-10. Best-hit orthologs were identified and reciprocal BLAST analysis was performed against the *D. melanogaster* genome (obtained from http://www.flybase.org) to verify putative orthologs. These orthologs are then mapped to the differential gene expression dataset to obtain differential gene expression values.

**Tardigrade desiccation and quantification for trehalose and CAHS quantification**. *H. exemplaris* specimens were starved to avoid contamination from food sources and desiccated according to our established protocol[7,45]. Starved dried tardigrades were resuspended in 200 μL of spring water and 1:100 dilution yielding 10 μL of diluted tardigrades was plated on a 35 mm × 50 mm petri dish. Tardigrades in this aliquot were counted under a dissecting microscope. The estimate of the total number of tardigrades was calculated by multiplying the average of the counted tardigrades by the dilution factor (1:100).

Resuspended tardigrades transferred to a 1.5 ml centrifuge tube and flash-frozen in liquid nitrogen. Specimens were thawed and homogenized with a plastic pestle. Flash freezing, thawing, and homogenization was performed 6 times to ensure complete extraction of protectants. The sample was centrifuged at 10,000 g for 5 min and 20 μL of the supernatant was stored to measure trehalose concentration. The remaining volume of the sample was boiled for 10 min to precipitate all non-heat soluble proteins. The sample was then centrifuged at 10,000 g for 5 min and the supernatant, containing heat soluble proteins was removed and quantification of putative CAHS proteins was performed immediately.

**Quantification of putative CAHS proteins**. The heat-soluble supernatant from homogenized tardigrades was run on a Criterion TGX 4–20% SDS-page gel (BioRad Cat #5671094) along with a concentration gradient of BSA standards. Putative CAHS proteins were indicated at ~25.4 kDa, indicative of the molecular weight of CAHS D. ImageJ was used to quantify band intensities of the BSA standards and putative CAHS protein. A standard curve was constructed using known BSA standards and their band intensities. The slope of the line of best fit for this standard curve was found to be 0.034 with a $R^2$ value of 0.9986. Using this equation and the band intensity of our putative CAHS protein, an approximation of CAHS concentration was obtained.

**Quantification of Trehalose**. Trehalose concentration was quantified using the Megazyme Trehalose Assay Kit (K-TREH 01/20) according to the manufacturer's instructions. 180 μL of Alkaline borohydride (10 mg/mL sodium borohydride, 50 mM sodium hydroxide [Sigma-Aldrich]) was added to the sample to remove any reducing sugars. Controls of known sucrose and trehalose concentrations were run alongside this sample. In a well, 200 μL of sample depleted of reducing sugars was combined with 20 μL of solution 1, 10 μL of solution 2 (NADP⁺/ATP), and 2 μL of suspension solution 3 and was incubated for 5 min. 2 μL of Trehalase was then added to catalyze the conversion of trehalose to glucose. Accumulated NADPH was measured at 340 nm using a plate reader (Spark 20 M, Tecan, Männedorf, Switzerland) at 10 s intervals for 5 min. The concentration of NADPH is stoichiometric with D-glucose, equating to twice the amount of trehalose.

**Cloning of CAHS D**. CAHS D gblock (Integrated DNA Technologies) codon optimized for expression in *E. coli* was cloned into pET28b expression vector using Gibson assembly. Sanger sequencing was used to confirm the full incorporation of CAHS D into pET28b.

**Expression of CAHS D**. Expression constructs were transformed in BL21 (DE3) *E. coli* (New England Biolabs) and plated on LB agar plates containing 50 μg/mL kanamycin. Large-scale expression was performed in 1 L LB/kanamycin cultures, shaken at 37 °C until an optical density of 0.6 was obtained. Expression was induced using 1 mM IPTG for 4 h. Cells were harvested by centrifugation at 4000 g

at 4 °C for 30 min. Cell pellets were resuspended in 5 mL of resuspension buffer (20 mM tris, pH 7.5), and 30 μL protease inhibitor (Sigma-Aldrich, St. Louis, MO). Pellets were stored at −80 °C.

**Purification of CAHS D**. Purification largely follows the methods in Piszkiewicz et al.[27]. Pellets were allowed to thaw at room temperature and heat lysis was performed by boiling for 10 min. All insoluble components were removed via centrifugation at 5000 g at 10 °C for 30 min. The supernatant was sterile filtered with 0.45 μm and 0.22 μm syringe filters (Foxx Life Sciences, Salem, NH). The filtered lysate was diluted until a final volume of 60 mL was obtained using purification buffer UA (8 M Urea, 50 mM sodium acetate [Acros Organics, Carlsbad, CA], pH 4). The protein was then purified using a cation exchange HiPrep SP HP 16/10 column (Cytiva, Marlborough, MA) on an AKTA Pure 25 L (Cytiva) system, controlled using the UNICORN 7 Workstation pure-BP-exp (Cytiva). Protein was eluted using a step method of 40% UB (8 M Urea, 50 mM sodium acetate, and 1 M NaCl, pH 4).

Fractions of eluent were confirmed using SDS-PAGE and were pooled for dialysis in 3.5 kDa MWCO dialysis tubing (SpectraPor 3 Dialysis Membrane, Sigma-Aldrich). Identified fractions were dialyzed at room temperature for 4 h against 20 mM sodium phosphate at pH 7.0. This was followed by six rounds of dialysis in Milli-Q water (18.2 MΩcm). Samples were then quantified fluorometrically (Qubit4 Fluorometer, Invitrogen, Waltham, MA) and lyophilized (FreeZone 6, Labconco, Kansas City, MO) for 48 h. Purified, lyophilized protein was stored at −20 °C.

**LDH protection assay**. LDH desiccation protection assays were performed in triplicate using protocols from Boothby et al.[7,45]. Protectants were resuspended in a concentration range from 0.01 mg/mL to 20 mg/mL in 100 μL resuspension buffer (25 mM Tris, pH 7.0). L-LDH from rabbit muscle (Sigma-Aldrich Cat #10127230001) was added at 0.1 g/L. 50 μL of each sample was stored at 4 °C while the other half was desiccated for 18 h (OFP400, Thermo Fisher Scientific, Waltham, MA). All samples were brought to a volume of 250 μL with water. The enzyme/protectant mixture was added 1:10 to the assay buffer (100 mM Sodium Phosphate, 2 mM Sodium Pyruvate [Sigma-Aldrich], 1 mM NADH [Sigma-Aldrich], pH 6). Conversion to NAD + was measured by Enzyme kinetics at 340 nm using the UV-Vis function of NanodropOne (Thermo Fisher Scientific). The percent activity of LDH was determined by comparing the initial, linear reaction rate of each dehydrated and rehydrated sample to its corresponding unstressed (hydrated) control.

**LDH assay synergy assays**. Preparations of LDH desiccation protection mixing experiments were identical to the procedures above (LDH protection assay), aside from the preparation of protectant mixtures. Suboptimal concentrations of CAHS D and BSA were used for these mixing experiments. Proteins and disaccharides were resuspended at 2:1 molar ratio of the experimental target concentration in 100 μL resuspension buffer. 50 μL of each protectant at targeted ratio was combined in a 1.5 mL centrifuge tube. The procedure is completed according to the methods above.

**RNA integrity assay**. RNA encoding for enhanced green fluorescent protein (eGFP) was transcribed utilizing a MEGAscript™ T7 transcription kit and purified using a MEGAclear™ transcription clean-up kit. Translation was performed using nuclease-treated rabbit reticulocyte lysate, purchased from Promega. RNAse-free 1 M trehalose solution was purchased from Ambion.

3 mg of RNA was mixed with 1 mL of 88.2 mM CAHS D and 2 mL of either 35.5 mM, 355 mM, or 3.55 mM Trehalose to obtain a 1:0.8, 1:8, or 1:80 mixture of CAHS D:Trehalose, respectively. Each sample was then brought to 6 mL using nuclease-free water and aliquoted into three 2 mL samples, before being vacuum dried for 3 h and placed into a desiccation chamber at 60 °C for 7 days. Samples were then removed from the desiccation chamber and allowed to cool to room temperature. Samples were rehydrated and assembled into 25 mL translation reactions following the manufacturer's instructions and incubated at 30 °C in a thermocycler for 90 min. After translation, the fluorescence of eGFP for each sample was measured utilizing a 384-well plate on a Tecan CytoSpark plate reader.

**Yeast survival assay**. Cells were grown to mid-exponential phase (OD < 0.5) in selective media SC-Uracil (CAHS genes are expressed under the constitutive promoter *TDH3* in a p416 (Uracil selection) plasmid; empty vector is used as control). Cells were then transferred to selective media with trehalose (SC-Ura + 2% trehalose) for 1 h. Following induction, ~10E7 cells were withdrawn from liquid cultures, washed twice in water, and then brought to a final volume of 1 mL. Non-desiccated controls were assessed for viability by counting colony forming units (CFUs) formed from 200 μl of resuspended cells. 200 μL aliquots were transferred to a 96-well tissue culture plate, centrifuged, and water was removed without disturbing the cell pellet. Cells were allowed to desiccate at 23 °C with constant 60% relative humidity (RH), for 48 h. After 48 h, cells were rehydrated and assessed for viability by CFUs.

**Reporting Summary**. Further information on research design is available in the Nature Research Reporting Summary linked to this article.

## Data availability

All data generated or analyzed during this study are included in this published article (Data S1.zip).

## Code availability

All custom code used in this study are included in this published article (Data S2.zip).

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

## Acknowledgements

This work was supported by DARPA (W922NF-20-2-0137) and NSF (IntBio 2128069) to TCB. In addition, this work was made possible in part through support from an Institutional Development Award (IDeA) from the National Institute of General Medical Sciences of the National Institutes of Health (Grant # 2P20GM103432). The authors are thankful to Drs. Silvia Sanchez Martinez and Shahar Sukenik as well as members of the Boothby Lab for discussions and reading of this manuscript. We thank members of the Water and Life Interface Institute (WALII), supported by NSF DBI grant # 2213983, for helpful discussions. Bioinformatics analysis was done using the Advanced Research Computing Center's Teton Computing Environment at the University of Wyoming (https://doi.org/10.15786/M2FY47).

## Author contributions

K.N., H.T., T.G., S.K. and T.C.B. participated in experimental design and preparation of the manuscript and figures. K.N. performed metabolomic and biochemical analysis. K.N. and S.K. performed enzyme protection/synergy experiments. T.G. performed and

analyzed RNA experiments. H.T. performed and analyzed yeast experiments. The authors have read and approved the final manuscript.

## Competing interests

The authors declare no competing interests.
