## [Peer Review File · Communications Biology]

Reviewers' comments:

Reviewer #1 (Remarks to the Author):

Summary Statement

The research article by Nguyen et al. evaluates the role of cytoplasmic abundant heat soluble (CAHS) proteins and trehalose for desiccation tolerance in the tardigrade *H. exemplaris*. Metabolomics analysis found that trehalose levels were five times higher in preconditioned and desiccated tardigrades compared to fully hydrated animals. Furthermore, differences in the metabolome, including the levels of trehalose and UDP-glucose, were predictive of the anhydrobiotic state. The authors suggest that trehalose accumulates in tardigrades by preventing its breakdown mediated by trehalase, but trehalase activity is not being assessed. The changes in trehalase transcripts show lower abundance, but the statement that the regulatory mechanism of increasing cellular trehalose levels is based on a reduced trehalase activity is premature. Another concern is that the amount of trehalose measured might be impacted by the food source or the gut microbiome. Are the authors confident that using whole well-fed animals means all measured metabolites are cellular and unrelated to gut content and the microbiome? Assays measuring lactate dehydrogenase activity after desiccation demonstrated that CAHS proteins plus trehalose conferred synergistic protection to the enzyme lactate dehydrogenase at some protein-to-trehalose ratios. These results were strengthened by an *in vivo* yeast assay. Overall, the paper is well written and provides insights into the role of CAHS proteins and trehalose in conferring desiccation tolerance. However, several concerns need to be addressed and are detailed below.

Results

Overall

1. Lines 93 – 95. Please provide a rationale for not performing metabolomics on preconditioned but not desiccated tardigrades. In this context, the percentages of animals surviving preconditioning and desiccation after preconditioning should be reported.
2. Line 122. Remove 'analysis' from 'RFA analysis' since RFA stands for 'random forest analysis'.
3. Fig 1E. Line 150 – 157. The figure shows that the mean-decrease-accuracy for PFOA is higher than for trehalose and UDP-glucose. Please explain the significance behind this result and the reason a perfluorochemical could have any role in the predictive accuracy of determining the anhydrobiotic state. I encourage the author to rationally assess which data are to be reported from the pool of data obtained from untargeted metabolomics.
4. Figure 1G. Why is the scaled intensity of trehalose in the hydrated tardigrade ~ 0.5 ? Shouldn't this be standardized to 1? What does a 'scaled intensity' of 1 represent? Why did the authors not measure trehalose concentrations in hydrated animals as was performed for anhydrobiotic animals reported in Table 1? This approach would allow for a direct comparison based on concentrations.
5. Lines 179 – 215. These paragraphs summarize results from previous studies and discuss results from Fig. 1. Consider omitting or moving some of this text to the introduction or discussion sections. Alternatively, the authors could restructure the paper to have a combined 'results and discussion' section.
6. Table 1. Using the data provided in the table, we calculated 16.3 fmol of trehalose per tardigrade which would change the molar ratio of trehalose to CAHS protein. We believe pmol should be altered to fmol in the table, but please double-check your calculations.

7. Table 1. Please refer to Fig. S2 when presenting/discussing the measured CAHS protein concentration.
8. Line 227. ... 1.6 µg... (not 1.6 ug).
9. Line 229. "or ~16 fmol per tardigrade (Table 1)." In the table, this value is reported as 1.96 pmol of protectant per tardigrade, equivalent to 1960 fmol per tardigrade. The table or text has an error. Please confirm your calculations.
10. Line 241 – 243. The statement in these lines contradicts the information in lines 236 – 237 that all key enzymes of trehalose metabolism were identified. We feel that Tsp1 and TPP are critical enzymes in trehalose synthesis. Could there be an alternative pathway for tardigrades to synthesize trehalose, considering there are five different pathways of trehalose synthesis in bacteria? Furthermore, how high is the trehalose content of their algal food source? Could some of the reported trehalose be from the algae and bacteria in their gut?
11. Line 272 – 295. These lines fit more into an introduction or discussion section, and you may consider restructuring the manuscript.
12. Line 302 – 303. The authors state that this analysis estimates how much CAHS protein and trehalose are co-enriched with trehalose during desiccation. Yet, their data provided no data on CAHS protein levels before preconditioning.
13. Line 304 – 305. Please compare your values of CAHS proteins in the text and table. The table states 0.24 p mol CAHS, while the text states 1.97 fmol.
14. Line 326. ... since BSA should: i)... (Add colon before list)
15. Lines 328 – 329. Remove "(cows, from which BSA is derived, are not anhydrobiotic)."
16. Line 339. Refers to supplemental figure 4. However, the supplemental Fig. 3 was not introduced at this point. Please reorder the figures.
17. Line 345. Revisit the discrepancies found in the empirically determined biological ratios from Table 1 and text. Furthermore, since concentrations in 3A are given in mg/mL, would you please estimate the cytoplasmic concentration of CAHS proteins in mg/mL? We also encourage the author to add a column to Table 1, which provides the concentrations of trehalose and CAHS protein in mg/mL cytoplasmic water as reported for the in vitro experiments.
18. Figure 4. Please provide the p values of your analysis. Furthermore, these set of data should be analyzed by an ANOVA test. Finally, the legend is misleading since the black bar indicates trehalose, yet sucrose was used in some experiments. Clearly state which sugar and protein were used in each experiment represented by a bar in Fig. 4. It is unclear why empty vector DNA was added to any control samples (red square).
19. Figure 4C and 4D. Why is a bar showing the percentage of protection provided by CAHS alone shown in 4C, 4D? In these subfigures the protection of BSA alone should be shown since you are evaluating if there is a synergistic effect between these compounds.
20. Line 403. Please change "east" to "Yeast".
21. Line 418. Please provide units for the trehalose concentration. I am assuming it's 2% w/v.
22. Figure 5B – 5F. All figures indicate a significant difference between the vector control and CAHS

protein groups. However, visually they appear to be both 0. Please provide the raw data in your supplemental files for the colony counts.

Discussion

23. The discussion section reads like a conclusion section since the discussion section should be a comparison to previous literature (no sources cited in the discussion section). Consider changing the results section to a results/discussion section and the discussion section to a conclusion section.

Methods

24. Line 468. Is the metabolite composition of the food source known since animals were not utilized in a starved state? How much of the trehalose could be due to the algae or microbiome residing in the digestive track?

25. Line 515 – 523. Figure S2. Please provide more background about the SDS PAGE used to quantify CAHS protein concentrations. Furthermore, more than one sample should be used to judge the CAHS protein concentration in these animals.

26. Line 571. LDH Protection Assay. Please provide the product code for the used LDH and state the species and formulation of the product received from Sigma-Aldrich.

27. Lines 581 – 582. How was the percent LDH protection calculated? If the enzyme was allowed to react to completion, all the NADH would have been converted to NAD⁺. To determine the percent of protection, you need to calculate the percentage of LDH activity that was recovered. You need to express LDH activity (a rate: dABS/time) recovered after rehydration as a percentage of the LDH activity (dABS/time) before desiccation and rehydration. You cannot simply use the ratio of the NAD⁺ absorbance at a single time point.

Reviewer #2 (Remarks to the Author):

Nguyen et al. provide a compelling report on the role of trehalose as a required co-solute to drive the protective function of CAHS proteins during tardigrade desiccation. The role of trehalose has been abundantly described in desiccation-tolerant organisms, yet, tardigrades seemed to be an exception to this rule. Trehalose is only found at low levels or is even absent in the assayed tardigrades. This led to the assumption that trehalose had a negligible role in tardigrade desiccation tolerance. In this study, the authors show that a model tardigrade does consistently accumulate low levels of trehalose in desiccation-tolerant conditions, together with a host of other metabolites. Since previous work by the authors described a profound role for disordered CAHS proteins, the authors assessed whether both these protectants could work synergistically. Compellingly, the authors find that even low levels of trehalose can dramatically exacerbate the protective role of CAHS in both in vitro and in vivo assays. This study for the first time highlights the importance of trehalose to desiccation tolerance in tardigrades and uncovers a new interesting aspect of CAHS biology. Importantly, the findings of this study extend well beyond tardigrade and desiccation biology. For decades the motto of the intrinsically disordered protein (IDP) field is that IDPs are exquisitely sensitive to the (bio)chemical environment. Yet, how this sensing capability has any functional biological effects remains largely unexplored. This study blows this area of research wide open and provides a rare example of the functional effect of IDP-metabolite interactions. Given this broad impact, the rigorous experimental design, and overall quality of the work, I can only give my strongest endorsement for publication of this article in *Communications Biology*. This Reviewer has only found two small typos, but does not believe that further experiments are required to support the findings of the presented work.

Typos:

Figure S4: Highly enriched metabolites in desiccated tardigrades DO not confer protection alone.

Line 403: nth1Δ + TDH3pr-Agt1 background. (Fig. 5A). YEAST with this background lack Nth1, an

Reviewer #3 (Remarks to the Author):

The paper "Trehalose and tardigrade CAHS proteins work synergistically to promote desiccation tolerance" deals with the thorough metabolomic analysis of hydrated and dried tardigrades pertaining to *Hypsibius exemplaris* species. Even though trehalose levels do not change during desiccation, the sugar's synergistic action with desiccation induced CAHS proteins is proved to be highly efficient in giving protection both in in vitro and in vivo experiments.

Conclusions are of interest in the community and can be interesting also for the wider field. Moreover, the statistical analyses and the information for reproducibility of the experiment are sound, clear and detailed.

The only caveat that I have, is the relative shortness of the discussion section. I think that more thorough analysis of these new data should be put forward. An idea could be also to examine possible implications on the possibility to confer desiccation tolerance to other cells/organisms.

Some minor corrections should be be addressed by the Authors:

1- Throughout the ms, unit measures are sometimes written adjacent to their values, while sometimes there is a space between them. Please uniform.

2- Throughout the ms, the letter 'u' is frequently used to replace the Greek letter 'mu', while sometimes the Greek letter is properly utilized (see f.e. page 7, line 15; page 16, lines 26 and 29). Please use the right notation in all parts of the ms.

3- Page 2, lines 4 and 7, please change " 'omics " into " -omics ".

4- Page 2, line 9, please change 'takes' into 'take'.

5- Page 2, line 32, please change "sugars" into "sugar's".

6- Page 3, line 3, please change "demonstrates" into "demonstrates".

7- Page 3, line 19, please change '1A' to '1B'.

8- Page 5, Figure 1, please add the abbreviations 'PCA' in section D, and 'RFA' and 'MDA' in section E, for allowing an easier reading.

9- Page 10, line 5 'H. exemplaris' should be in italics.

10- Page 14, line 6, 'est' should probably be replaced by 'Yeast'.

11- Page 17, line 5, the sentence starts with a '_', which should be deleted.

12- Page 17, line 5, reference number for Boothby et al., 2017 should be added (#7).

13- Page 19, line 12, reference number for Boothby et al., 2017 should be added (#7).

14- References 3, 4, 9, 20, 21, 24, 26, 31, 42, 45 (page 20-24) are incomplete. Issue numbers and page ranges should be added.

Nguyen *et al.*, point-by-point response to reviewers

We are grateful to all three reviewers for their encouragement and constructive criticism. Below we provide a point-by-point response for each of the concerns/comments made. In the document below, reviewer comments are left in regular font, while **our responses are shown in bold**.

Reviewers' comments:

Reviewer #1 (Remarks to the Author):

Summary Statement

The research article by Nguyen et al. evaluates the role of cytoplasmic abundant heat soluble (CAHS) proteins and trehalose for desiccation tolerance in the tardigrade *H. exemplaris*. Metabolomics analysis found that trehalose levels were five times higher in preconditioned and desiccated tardigrades compared to fully hydrated animals. Furthermore, differences in the metabolome, including the levels of trehalose and UDG-glucose, were predictive of the anhydrobiotic state.

The authors suggest that trehalose accumulates in tardigrades by preventing its breakdown mediated by trehalase, but trehalase activity is not being assessed. The changes in trehalase transcripts show lower abundance, but the statement that the regulatory mechanism of increasing cellular trehalose levels is based on a reduced trehalase activity is premature.

We agree with the reviewer and have softened the prose in our manuscript as well as added a statement that to fully support this theory future experiments looking at the enzyme activity of trehalases in drying tardigrades will need to be performed.

Another concern is that the amount of trehalose measured might be impacted by the food source or the gut microbiome. Are the authors confident that using whole well-fed animals means all measured metabolites are cellular and unrelated to gut content and the microbiome?

As noted below when the reviewer brings up this concern, we did starve our animals following previously established (cited) protocols. We apologize for not making this point clearer, since it is a very important one. We have added text to our methods and results section pointing out that both hydrated and dried animals were starved to reduce the possibility of contamination from algal food/microbial sources.

Assays measuring lactate dehydrogenase activity after desiccation demonstrated that CAHS proteins plus trehalose conferred synergistic protection to the enzyme lactate

dehydrogenase at some protein-to-trehalose ratios. These results were strengthened by an in vivo yeast assay. Overall, the paper is well written and provides insights into the role of CAHS proteins and trehalose in conferring desiccation tolerance. However, several concerns need to be addressed and are detailed below.

Results

Overall

1. Lines 93 – 95. Please provide a rationale for not performing metabolomics on preconditioned but not desiccated tardigrades. In this context, the percentages of animals surviving preconditioning and desiccation after preconditioning should be reported.

Preconditioned animals were not alongside preconditioned+dried and hydrated samples simply due to the large input of animals needed for 9 metabolomic replicated (needed for statistical robustness). For reference, it took us nearly two years to culture enough animals for the studies we did perform and while it would be fantastic to have added this additional condition, we felt the added time was too much. We have added prose in our manuscript to this effect.

As the reviewer suggests we have also added numbers and references for survival rates of preconditioned, preconditioned+dried, and non-preconditioned dried animals

2. Line 122. Remove 'analysis' from 'RFA analysis' since RFA stands for 'random forest analysis'.

Fixed this and other instances of the same error.

3. Fig 1E. Line 150 – 157. The figure shows that the mean-decrease-accuracy for PFOA is higher than for trehalose and UDP-glucose. Please explain the significance behind this result and the reason a perfluorochemical could have any role in the predictive accuracy of determining the anhydrobiotic state. I encourage the author to rationally assess which data are to be reported from the pool of data obtained from untargeted metabolomics.

We see the reviewer's point. We had originally thought to leave everything in our analysis to avoid any concerns of 'cherry-picking' our data. However, we agree that PFOA, a man-made compound, would not be produced by tardigrades and was likely introduced as a contaminant somewhere during the metabolomics pipeline. As the reviewer suggests, we have removed PFOA and the other obvious xenobiotics from our RFA analysis. We believe this will allow readers to more easily focus on metabolites that have a realistic potential of contributing to tardigrade desiccation tolerance. We have added text to our RFA method explaining this decision.

4. Figure 1G. Why is the scaled intensity of trehalose in the hydrated tardigrade ~0.5? Shouldn't this be standardized to 1? What does a 'scaled intensity' of 1 represent?

We apologize for the confusion in explaining how our metabolomic data was analyzed. First, for each of our replicates we obtained raw count numbers from mass-spec for all metabolites. These raw numbers were then normalized so the median raw score is equal to 1. These numbers were further normalized using BCA protein concentration (protein normalization) in each replicate and again the median set equal to 1. In short, the scaled intensity of trehalose in hydrated tardigrades is <1 because the amount of trehalose in these samples was lower than the median metabolite level in that condition. We agree with the reviewer that we *could* have set the intensity of trehalose in hydrated samples equal to 1, but in this case one would lose insight into how the levels of trehalose compare to other metabolites. We have added prose clarifying this to our materials and methods.

Why did the authors not measure trehalose concentrations in hydrated animals as was performed for anhydrobiotic animals reported in Table 1? This approach would allow for a direct comparison based on concentrations.

A direct measurement of trehalose in hydrated animals was not made because of the large input of animals needed to perform this analysis.

5. Lines 179 – 215. These paragraphs summarize results from previous studies and discuss results from Fig. 1. Consider omitting or moving some of this text to the introduction or discussion sections. Alternatively, the authors could restructure the paper to have a combined 'results and discussion' section.

We appreciate this suggestion and have combined our Results and Discussion section into one.

6. Table 1. Using the data provided in the table, we calculated 16.3 fmol of trehalose per tardigrade which would change the molar ratio of trehalose to CAHS protein. We believe pmol should be altered to fmol in the table, but please double-check your calculations.

We appreciate the reviewer pointing this discrepancy out. We have fixed this and other instances throughout the manuscript.

7. Table 1. Please refer to Fig. S2 when presenting/discussing the measured CAHS protein concentration.

We have added references to Fig. S2 when discussing measured CAHS protein concentrations.

8. Line 227. ... 1.6 µg... (not 1.6 ug).

We have fixed this and other instances throughout the manuscript.

9. Line 229. “or ~16 fmol per tardigrade (Table 1).” In the table, this value is reported as 1.96 pmol of protectant per tardigrade, equivalent to 1960 fmol per tardigrade. The table or text has an error. Please confirm your calculations.

Fixed

10. Line 241 – 243. The statement in these lines contradicts the information in lines 236 – 237 that all key enzymes of trehalose metabolism were identified. We feel that Tsp1 and TPP are critical enzymes in trehalose synthesis. Could there be an alternative pathway for tardigrades to synthesize trehalose, considering there are five different pathways of trehalose synthesis in bacteria?

We apologize for not being clearer in our prose. The *H. exemplaris* transcriptome does contain transcripts with high sequence similarity that reciprocally BLAST to Tps1 and TPP, however these transcripts are not expressed at detectable levels under hydrated or dry conditions. Interestingly, they are expressed under freezing conditions – and though interesting we feel following up on this point is beyond the scope of this paper. We have added to and edited our prose to make these points clearer.

Furthermore, how high is the trehalose content of their algal food source? Could some of the reported trehalose be from the algae and bacteria in their gut?

Again, our apologies. Our methods section could have been more explicit that all our samples were washed and starved prior to desiccation/sample preparation. This is our standard practice, but we merely cited our old papers and methods rather than saying this explicitly. We appreciate the reviewer pointing this out, since it is a very important point that our samples we washed and starved to reduce algal contamination. We have clarified this point in our results/discussion as well as methods section.

11. Line 272 – 295. These lines fit more into an introduction or discussion section, and you may consider restructuring the manuscript.

We have combined our results and discussion section into one.

12. Line 302 – 303. The authors state that this analysis estimates how much CAHS protein and trehalose are co-enriched with trehalose during desiccation. Yet, their data provided no data on CAHS protein levels before preconditioning.

The reviewer is 100% correct. This was a poor choice of words on our part. We have amended our prose to reflect that this data provides an estimate of the ratio

of CAHS protein and trehalose in dry *H. exemplaris* specimens – but says nothing about enrichment of these molecules.

13. Line 304 – 305. Please compare your values of CAHS proteins in the text and table. The table states 0.24 p mol CAHS, while the text states 1.97 fmol.

Fixed.

14. Line 326. ... since BSA should: i)... (Add colon before list)

A colon has been added.

15. Lines 328 – 329. Remove “(cows, from which BSA is derived, are not anhydrobiotic).”

Removed

16. Line 339. Refers to supplemental figure 4. However, the supplemental Fig. 3 was not introduced at this point. Please reorder the figures.

Figures S3 and S4 have been reordered so that their numbering reflects the order in which they are introduced in the main text of our manuscript.

17. Line 345. Revisit the discrepancies found in the empirically determined biological ratios from Table 1 and text. Furthermore, since concentrations in 3A are given in mg/mL, would you please estimate the cytoplasmic concentration of CAHS proteins in mg/mL? We also encourage the author to add a column to Table 1, which provides the concentrations of trehalose and CAHS protein in mg/mL cytoplasmic water as reported for the in vitro experiments.

We have fixed the discrepancies between Table 1 and values reported in text.

We have also added a column to Table 1 with the approximate concentration in mg/ml of trehalose and CAHS protein in dry tardigrades.

18. Figure 4. Please provide the p values of your analysis. Furthermore, these set of data should be analyzed by an ANOVA test. Finally, the legend is misleading since the black bar indicates trehalose, yet sucrose was used in some experiments. Clearly state which sugar and protein were used in each experiment represented by a bar in Fig. 4. It is unclear why empty vector DNA was added to any control samples (red square).

We have added p-values of our analysis derived from an ANOVA test.

We have also clarified the figure legend to indicate that the black bar represents the indicated sugar (sucrose or trehalose)

.

We apologize for the typo, empty vector was not added to these samples. “Empty vector” samples have been updated to “Control” samples, where no additive or protectant was supplied.

19. Figure 4C and 4D. Why is a bar showing the percentage of protection provided by CAHS alone shown in 4C, 4D? In these subfigures the protection of BSA alone should be shown since you are evaluating if there is a synergistic effect between these compounds.

We have fixed the typos in this figure.

20. Line 403. Please change “east’ to “Yeast”.

Fixed

21. Line 418. Please provide units for the trehalose concentration. I am assuming it’s 2% w/v.

The reviewer is correct that the units should be w/v and we have added the units to this statement.

22. Figure 5B – 5F. All figures indicate a significant difference between the vector control and CAHS protein groups. However, visually they appear to be both 0. Please provide the raw data in your supplemental files for the colony counts.

Raw data with the percent survival for vector control and CAHS proteins has been provided in supplemental files.

Discussion

23. The discussion section reads like a conclusion section since the discussion section should be a comparison to previous literature (no sources cited in the discussion section). Consider changing the results section to a results/discussion section and the discussion section to a conclusion section.

We appreciate this suggestion and have combined our results and discussion section.

Methods

24. Line 468. Is the metabolite composition of the food source known since animals were not utilized in a starved state? How much of the trehalose could be due to the algae or microbiome residing in the digestive track?

Our apologies. As noted above, our animals were starved. While we referenced methods from our previous papers that used starvation prior to drying, we did not

make it explicitly clear that both hydrated and dry animals were starved prior to their use. We did this to reduce the possibility of contamination of this sort. We have added text to both the results and discussion as well as the methods section of our manuscript to clarify this point.

25. Line 515 – 523. Figure S2. Please provide more background about the SDS PAGE used to quantify CAHS protein concentrations. Furthermore, more than one sample should be used to judge the CAHS protein concentration in these animals.

Additional details on the SDS PAGE gels and procedure used have been added to our methods section. We agree that having additional replicates here would be fantastic, however the number of tardigrades needed for these measurements is extremely high. Not wanting to overstate things or rely too heavily on a single measurement, we do point out in our manuscript that this number is a broad estimate.

26. Line 571. LDH Protection Assay. Please provide the product code for the used LDH and state the species and formulation of the product received from Sigma-Aldrich.

We have provided the requested information in our methods section.

27. Lines 581 – 582. How was the percent LDH protection calculated? If the enzyme was allowed to react to completion, all the NADH would have been converted to NAD⁺. To determine the percent of protection, you need to calculate the percentage of LDH activity that was recovered. You need to express LDH activity (a rate: dABS/time) recovered after rehydration as a percentage of the LDH activity (dABS/time) before desiccation and rehydration. You cannot simply use the ratio of the NAD⁺ absorbance at a single time point.

Yes, the reviewer is correct, and our wording was poor. We performed these measurements as we have done previous (e.g., in Boothby *et al.*, 2017) by comparing the LDH activity in the rehydrated sample to the rate of its corresponding hydrated control. We have added text clarifying this to our methods section.

Reviewer #2 (Remarks to the Author):

Nguyen et al. provide a compelling report on the role of trehalose as a required co-solute to drive the protective function of CAHS proteins during tardigrade desiccation. The role of trehalose has been abundantly described in desiccation-tolerant organisms, yet, tardigrades seemed to be an exception to this rule. Trehalose is only found at low levels or is even absent in the assayed tardigrades. This led to the assumption that trehalose had a negligible role in tardigrade desiccation tolerance. In this study, the authors show that a model tardigrade does consistently accumulate low levels of trehalose in desiccation-tolerant conditions, together with a host of other metabolites. Since previous work by the authors described a profound role for disordered CAHS

proteins, the authors assessed whether both these protectants could work synergistically. Compellingly, the authors find that even low levels of trehalose can dramatically exacerbate the protective role of CAHS in both in vitro and in vivo assays. This study for the first time highlights the importance of trehalose to desiccation tolerance in tardigrades and uncovers a new interesting aspect of CAHS biology. Importantly, the findings of this study extend well beyond tardigrade and desiccation biology. For decades the motto of the intrinsically disordered protein (IDP) field is that IDPs are exquisitely sensitive to the (bio)chemical environment. Yet, how this sensing capability has any functional biological effects remains largely unexplored. This study blows this area of research wide open and provides a rare example of the functional effect of IDP-metabolite interactions. Given this broad impact, the rigorous experimental design, and overall quality of the work, I can only give my strongest endorsement for publication of this article in Communications Biology. This Reviewer has only found two small typos, but does not believe that further experiments are required to support the findings of the presented work.

Typos:

Figure S4: Highly enriched metabolites in desiccated tardigrades DO not confer protection alone.

Fixed

Line 403: nth1 Δ + TDH3pr-Agt1 background. (Fig. 5A). YEAST with this background lack Nth1, an

Fixed

Reviewer #3 (Remarks to the Author):

The paper "Trehalose and tardigrade CAHS proteins work synergistically to promote desiccation tolerance" deals with the thorough metabolomic analysis of hydrated and dried tardigrades pertaining to *Hypsibius exemplaris* species. Even though trehalose levels do not change during desiccation, the sugar's synergistic action with desiccation induced CAHS proteins is proved to be highly efficient in giving protection both in in vitro and in vivo experiments.

Conclusions are of interest in the community and can be interesting also for the wider field. Moreover, the statistical analyses and the information for reproducibility of the experiment are sound, clear and detailed.

The only caveat that I have, is the relative shortness of the discussion section.

I think that more thorough analysis of these new data should be put forward. An idea could be also to examine possible implications on the possibility to confer desiccation tolerance to other cells/organisms.

We appreciate the reviewers concern and suggestion. We have expanded our discussion, specifically mentioning implications for conferring desiccation tolerance to other organisms or cells.

Some minor corrections should be addressed by the Authors:

1- Throughout the ms, unit measures are sometimes written adjacent to their values, while sometimes there is a space between them. Please uniform.

We have updated the manuscript to get rid of this inconsistency.

2- Throughout the ms, the letter 'u' is frequently used to replace the Greek letter 'mu', while sometimes the Greek letter is properly utilized (see f.e. page 7, line 15; page 16, lines 26 and 29). Please use the right notation in all parts of the ms.

We have changed all instances where the Greek 'mu' should have been used instead of 'u.'

3- Page 2, lines 4 and 7, please change " 'omics " into " -omics ".

Fixed

4- Page 2, line 9, please change 'takes' into 'take'.

Fixed

5- Page 2, line 32, please change "sugars" into "sugar's".

Fixed

6- Page 3, line 3, please change "demonstrates" into "demonstrates".

Fixed

7- Page 3, line 19, please change '1A' to '1B'.

Fixed

8- Page 5, Figure 1, please add the abbreviations 'PCA' in section D, and 'RFA' and 'MDA' in section E, for allowing an easier reading.

We have used the terms PCA, RFA, and MDA in Figure 1 to help with readability.

9- Page 10, line 5 'H. exemplaris' should be in italics.

Fixed

10- Page 14, line 6, 'est' should probably be replaced by 'Yeast'.

Fixed

11- Page 17, line 5, the sentence starts with a '_', which should be deleted.

Fixed

12- Page 17, line 5, reference number for Boothby et al., 2017 should be added (#7).

Fixed

13- Page 19, line 12, reference number for Boothby et al., 2017 should be added (#7).

Fixed

14- References 3, 4, 9, 20, 21, 24, 26, 31, 42, 45 (page 20-24) are incomplete. Issue numbers and page ranges should be added.

Fixed

REVIEWERS' COMMENTS:

Reviewer #1 (Remarks to the Author):

The authors have addressed all my concerns.